# Structural basis of Stu2 recruitment to yeast kinetochores

Jacob A Zahm[1†], Michael G Stewart[2†], Joseph S Carrier[2], Stephen C Harrison[1]*, Matthew P Miller[2]*

[1]Department of Biological Chemistry and Molecular Pharmacology, Harvard Medical School, and Howard Hughes Medical Institute, Boston, United States; [2]Department of Biochemistry, University of Utah School of Medicine, Salt Lake City, United States

**Abstract** Chromosome segregation during cell division requires engagement of kinetochores of sister chromatids with microtubules emanating from opposite poles. As the corresponding microtubules shorten, these 'bioriented' sister kinetochores experience tension-dependent stabilization of microtubule attachments. The yeast XMAP215 family member and microtubule polymerase, Stu2, associates with kinetochores and contributes to tension-dependent stabilization in vitro. We show here that a C-terminal segment of Stu2 binds the four-way junction of the Ndc80 complex (Ndc80c) and that residues conserved both in yeast Stu2 orthologs and in their metazoan counterparts make specific contacts with Ndc80 and Spc24. Mutations that perturb this interaction prevent association of Stu2 with kinetochores, impair cell viability, produce biorientation defects, and delay cell cycle progression. Ectopic tethering of the mutant Stu2 species to the Ndc80c junction restores wild-type function in vivo. These findings show that the role of Stu2 in tension-sensing depends on its association with kinetochores by binding with Ndc80c.

*For correspondence:
harrison@crystal.harvard.edu (SCH);
matthew.miller@biochem.utah.edu (MPM)

[†]These authors contributed equally to this work

Competing interests: The authors declare that no competing interests exist.

## Introduction

Equal partitioning of duplicated chromosomes during cell division preserves integrity of the genome in each of the two daughter cells. 'Bioriented attachment' of sister chromatids to opposite poles of the mitotic spindle in turn ensures that when a cell enters anaphase, each pair of sister chromatids segregates accurately (reviewed in *Cheeseman, 2014*). When correctly bioriented, a pair of cohesin-linked sister chromatids will be under tension, from forces exerted by the shortening of opposing microtubules and transmitted through the kinetochores that connect spindle microtubules to chromosome centromeres. 'Tension-sensing' and correction of erroneous attachments are thus critical mediators of genome integrity.

The Aurora B kinase (Ipl1 in budding yeast) is the immediate agent of error correction (*Biggins and Murray, 2001*; *Biggins et al., 1999*; *Cheeseman et al., 2002*; *DeLuca et al., 2006*; *Hauf et al., 2003*; *Tanaka et al., 2002*). In the absence of tension, Ipl1 phosphorylates Ndc80 (*DeLuca et al., 2006*; *Hauf et al., 2003*), an essential part of the microtubule-contacting apparatus of a yeast kinetochore, and several other kinetochore substrates, including critical targets within the Dam1 complex (*Biggins et al., 1999*; *Cheeseman et al., 2002*; *Kang et al., 2001*). Phosphorylation induces dissociation of a kinetochore from its associated microtubule, interrupting the incorrect attachment and allowing the attachment search to 'try again'. Ndc80 is part of a rod-like assembly, the Ndc80 complex, a heterotetramer composed of two coiled-coil heterodimers, joined end-to-end in parallel orientations (*Alushin et al., 2010*; *Ciferri et al., 2008*; *Wei et al., 2005*; *Wilson-Kubalek et al., 2008*). The globular domain at the N-terminal end of Ndc80:Nuf2 contacts microtubules; the globular domain at the C-terminal end of Spc24:Spc25 interacts, through two distinct adaptors, with the chromatin-proximal inner kinetochore. The Ipl1 substrate residues of Ndc80 are in

an N-terminal extension that contributes, along with a globular, calponin homology domain, to the microtubule interface (*DeLuca et al., 2006*; *Umbreit et al., 2012*).

A further component of the response to erroneous attachment is the plus-end microtubule polymerase, Stu2 (the yeast ortholog of XMAP215/ch-TOG in metazoans). Its polymerization activity depends on plus-end binding, through a set of N-terminal TOG domains (*Figure 1A*; *Al-Bassam et al., 2006*; *Ayaz et al., 2012*; *Ayaz et al., 2014*; *Fox et al., 2014*), but its kinetochore association is independent of microtubules (*Hsu and Toda, 2011*; *Kakui et al., 2013*; *Miller et al., 2016*; *Miller et al., 2019*; *Vasileva et al., 2017*; *Herman et al., 2020*). In single-molecule experiments in vitro, kinetochores under low tension detach more frequently from microtubules than under higher tension (*Akiyoshi et al., 2010*), but only when Stu2 is present (*Miller et al., 2016*). Cells lacking Stu2 have error-correction defects, including detached kinetochores, loss of biorientation, and spindle-assembly checkpoint-dependent cell cycle delay, although previous work has been unable to separate these phenotypes from pleiotropic effects arising from loss of Stu2 microtubule polymerase activity (*Al-Bassam et al., 2006*; *Humphrey et al., 2018*; *Kosco et al., 2001*; *Miller et al., 2016*; *Pearson et al., 2003*; *Severin et al., 2001*; *Wang and Huffaker, 1997*).

Stu2 associates with Ndc80c (*Miller et al., 2016*; *Miller et al., 2019*). We describe experiments here in which we found that a C-terminal segment (CTS) of Stu2 bound specifically at the junction of the Ndc80:Nuf2 and Spc24:Spc25 heterodimers and that conserved features of the four-chain overlap at that junction were important for the interaction. Mutations in Stu2 that disrupted this interaction, blocked association of Stu2 with isolated kinetochores in vitro, impaired cell viability, and caused defects in chromosome biorientation and in timely cell cycle progression. Ectopic tethering of the mutant Stu2 to Ndc80c rescued these defects, allowing us to assign them to the interaction with Ndc80c rather than to interactions with other binding partners, such as Bik1 and Spc72 (*Usui et al., 2003*; *Wolyniak et al., 2006*). We conclude that Stu2 stably associates with the Ndc80c four-way junction and that this association is critical for establishing bioriented kinetochore attachments and for maintaining cell viability. These findings are consistent with the notion that kinetochore-associated Stu2 is a central component of the tension-sensing mechanism in budding yeast.

## Results

### Interaction of a conserved Stu2 CTS with Ndc80c

To specify interactions that stabilize a Stu2:Ndc80c complex, we determined which regions of Stu2 are both necessary and sufficient for binding to Ndc80c (*Figure 1A*). Initial pulldown experiments showed that recombinant full-length Stu2 bound directly to immobilized recombinant full-length Ndc80c (*Figure 1B*). Both the flexibility and elongated shape of Ndc80c would hinder structural studies of the full-length proteins. We instead used the 'dwarf' version of Ndc80c (Ndc80c$^{dwarf}$) in which the coiled-coil shaft between the globular heads has been shortened to facilitate crystallization (*Valverde et al., 2016*). In pulldown experiments, we found that Stu2 binds Ndc80c$^{dwarf}$ as efficiently as it does full-length Ndc80c (*Figure 1B*). Subsequent pulldown experiments used immobilized Ndc80c$^{dwarf}$ and a construct comprising the Stu2 dimeric coiled-coil domain followed by a glycine-serine linker, joined in turn to the Stu2 CTS or fragments of it (*Humphrey et al., 2018*; *Miller et al., 2019*; *Usui et al., 2003*; *Wang and Huffaker, 1997*). We found that a segment at the C-terminus of Stu2, most of which is conserved among budding and fission yeast and referred to here as the 'CTS' (*Figure 1A*), is required for binding Ndc80c. Deletion of either an N-terminal portion of the CTS (residues 857–867), a C-terminal portion of the CTS (residues 868–888), or the stretch following the CTS that lacks conservation (residues 880–888) resulted in loss of in vitro binding (*Figure 1C,D*). Fluorescence polarization binding experiments showed that the CTS binds dwarf Ndc80c with a dissociation constant of approximately 48 µM (*Figure 1E*). Together, these data show that the C-terminal 33 residues of Stu2 are sufficient for Ndc80c binding.

### Structure of Stu2 bound with Ndc80c$^{dwarf}$

We co-crystallized the Ndc80c$^{dwarf}$ with a 33-mer Stu2 peptide (Stu2 residues 855–888) and determined the structure of the complex by molecular replacement using the published Ndc80c$^{dwarf}$ structure as a search model (*Figure 2A*). Clearly defined density in the resulting map corresponded to residues 862–888; there was no density corresponding to the N-terminal seven residues (*Figure 2B*),

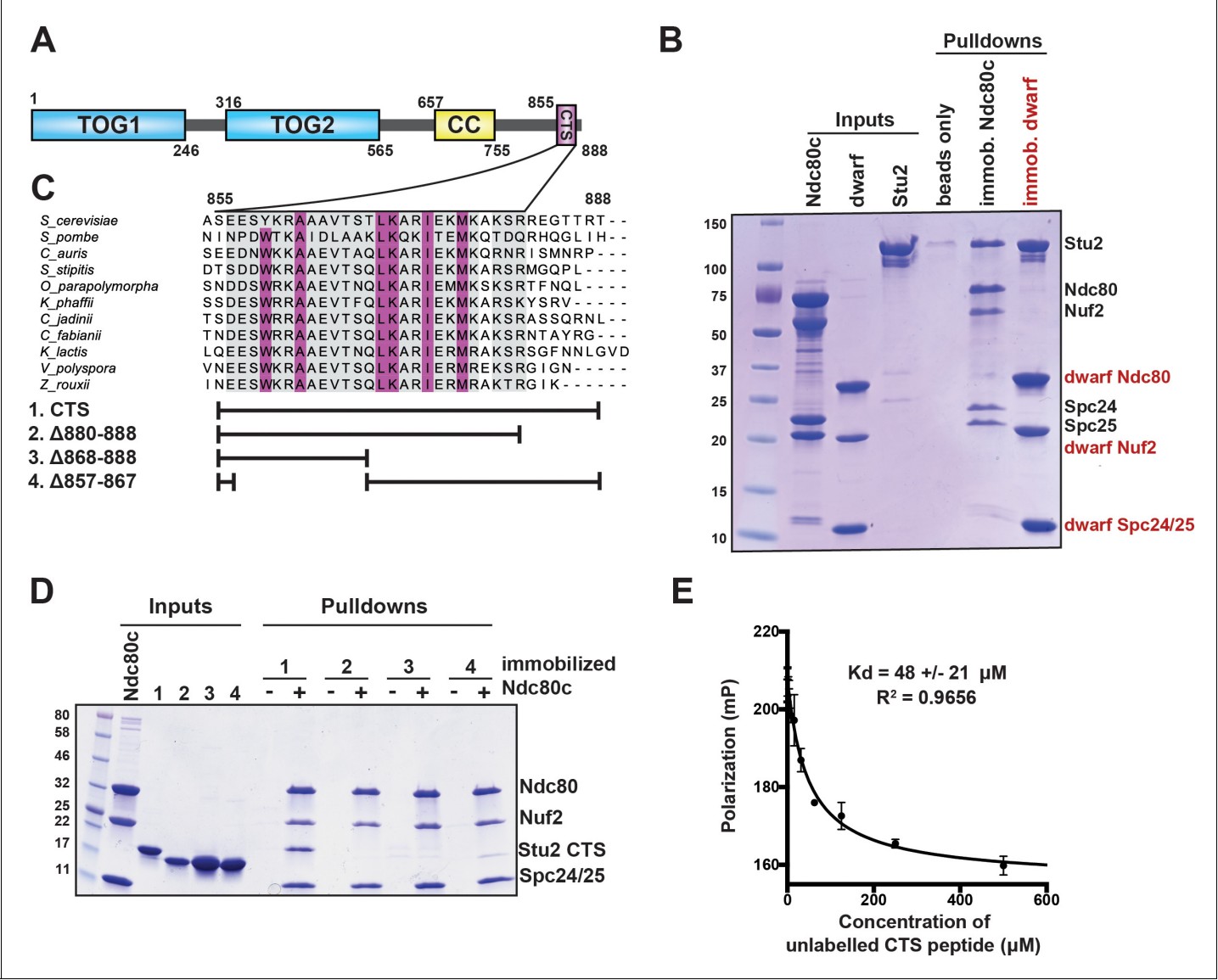

**Figure 1.** Binding of Stu2 C-terminal segment (CTS) with Ndc80c. (A) Schematic representation of the domain structure of Stu2, showing the tubulin-binding TOG domains, the dimeric coiled-coil, and the CTS. Molecular mass markers shown indicate molecular weight in kDa. (B) Association of full-length Ndc80c and Stu2. Dwarf and full-length Ndc80c were immobilized to saturation on Ni-NTA agarose and incubated with Stu2. After extensive washing, bound proteins were eluted with buffer containing 400 mM imidazole. (C) Multiple sequence alignment showing conservation of the CTS among budding and fission yeasts. Conservation calculated using T-Coffee Server (gray boxes) and percent identity calculated using Clustal Omega (purple boxes). The bars below the alignment correspond to the *Saccharomyces cerevisiae* sequence in the alignment and show the constructs used in the pulldown experiments in (D). The blank parts of each line represent deletions. (D) Binding of Ndc80c$^{dwarf}$ and Stu2 CTS. The Stu2 constructs used in this experiment consist of the Stu2 coiled-coil domain, followed by a glycine-serine linker, followed by the regions of the CTS indicated by the bars in (C). Ndc80c$^{dwarf}$ was immobilized on Co-NTA agarose and incubated with Stu2. After extensive washing, bound proteins were eluted with buffer containing 400 mM imidazole. (E) Affinity of Ndc80c$^{dwarf}$ and Stu2 CTS. Competition fluorescence polarization, showing displacement of Oregon-green labeled CTS peptide (50 nM) from Ndc80c$^{dwarf}$ (10 µM) with increasing concentrations of an unlabeled CTS peptide. Polarization (in milliP) is plotted against concentration of unlabeled peptide. Data fitted with a single-site saturation binding model implemented in GraphPad Prism 9.

and the C-terminal residues, 881–888, were part of a crystal-packing contact with the Nuf2 globular head of an Ndc80c symmetry mate (*Figure 2B*). This lattice interaction may account for the somewhat different unit-cell dimensions in our crystals from those of the unliganded Ndc80c$^{dwarf}$ in the same space group (*Valverde et al., 2016*), as well as for the nearly full occupancy but slightly higher local thermal parameters, as determined by refinement, for the low affinity peptide, which did not diffuse out during the short soak in cryopreservative (see Materials and methods). To resolve a

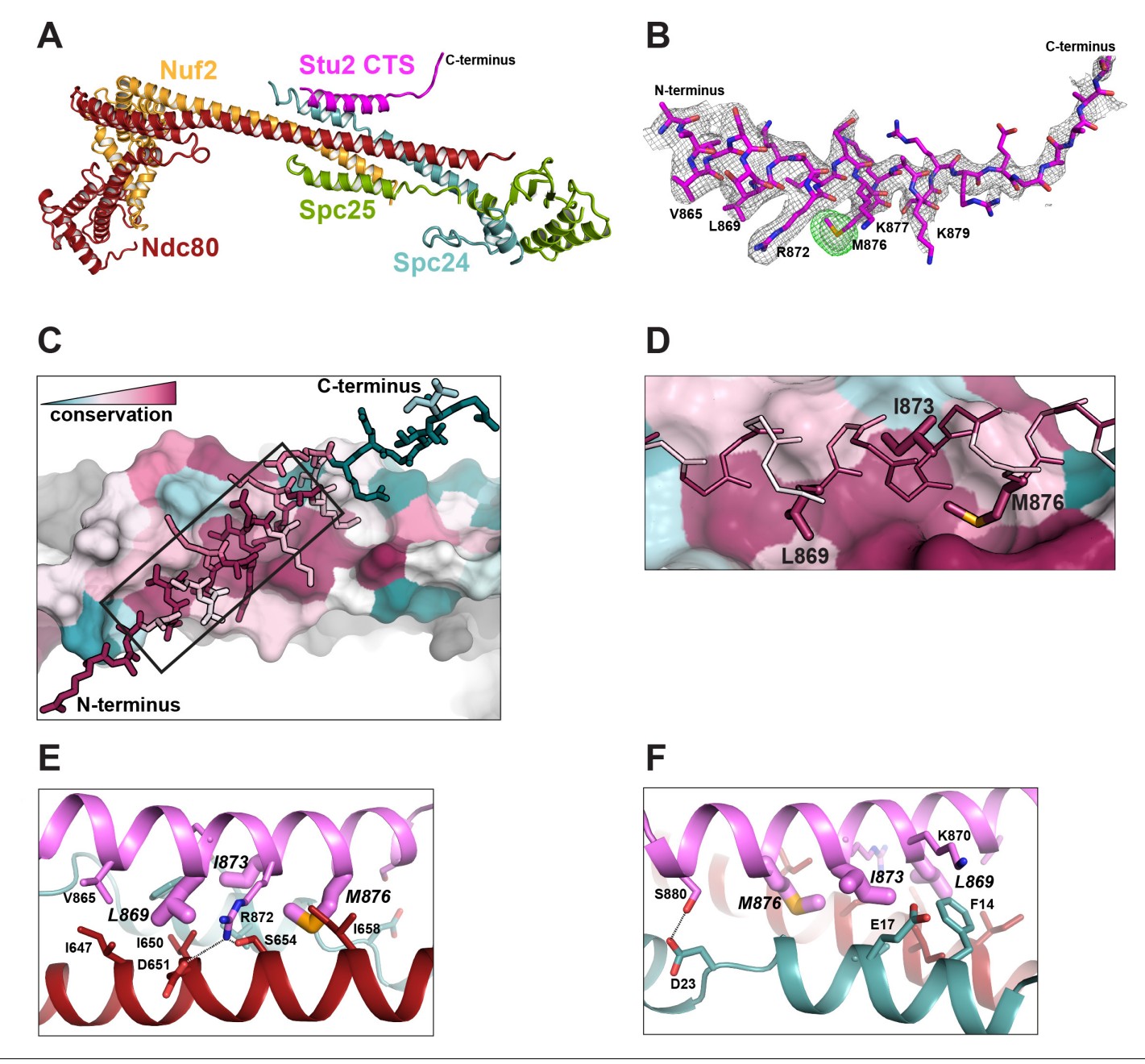

**Figure 2.** Structure of the Stu2 C-terminal segment (CTS) bound to the Ndc80c^dwarf. (A) Stu2 CTS (residues 855–888) bound to the Ndc80c^dwarf. The peptide binds at the four-way junction of Ndc80, Nuf2, Spc24, and Spc25 in a groove between Ndc80 and Spc24. (B) Model of the Stu2 peptide built into the 2Fo-Fc map (gray mesh); anomalous difference map (green mesh), contoured at 8σ, showing the position of SeMet in the peptide. (C) Conservation of residues at the contacts of Stu2 CTS, Ncd80, and Spc24, shaded from red (conserved) to blue (variable). Nuf2 and Spc25 are in gray. Ndc80c components in surface representation; Su2 in stick representation. The N- and C-termini of the Stu2 peptide are indicated. The expanded region depicted in (D) is shown as a black box. (D) Expanded view of the Ndc80-Spc24 surface corresponding to the boxed region in panel C, showing conserved pockets for the three hydrophobic residues of Stu2 discussed in the text. Coloring as in (C). (E and F) Detailed views of the contacts between the Stu2 peptide and Ndc80 and Spc24.

potential ambiguity in the sequence register of the model, we determined the structure of a complex with a peptide containing selenomethionine in place of the methionine at position 876. A strong anomalous difference peak confirmed that we had chosen the register correctly (*Figure 2B*).

The peptide binds at the junction of the two heterodimeric subcomplexes, contributing a fifth α helix, and packs at about 30° to the main axis of the four overlapping helices from Ndc80, Nuf2, Spc24, and Spc25. Four conserved hydrophobic-residue side chains in the Stu2 CTS (V865, L869, I873, and M876) fit into hydrophobic pockets lined by conserved residues of Ndc80 and Spc24 (*Figure 2A–F*), and in a conserved polar contact, the side chain of Stu2 R872 inserts between D651 and S654 on Ndc80 (*Figure 2E*).

To confirm the correspondence of the Stu2:Ndc80c interface in the crystal structure with binding in solution, we made point mutations in Stu2 and carried out pulldowns with immobilized Ndc80c$^{dwarf}$, using the same Stu2 constructs described above, except for the point mutations. For the binding experiments we mutated L869, I873, and M876 to alanine. The immobilized Ndc80c$^{dwarf}$ did not bind any mutant version of the Stu2 C-terminus (*Figure 3A*). These experiments show that the details of interaction between Stu2 and Ndc80c in the crystal lattice occur in solution as well.

## Kinetochore-associated Stu2 is required for cell viability

We generated strains containing point mutations at the positions tested in vitro, comprising three of the four conserved hydrophobic residues on Stu2 that contact Ndc80 and Spc24. For in vivo assays, we engineered an auxin-inducible degron (AID) in which addition of auxin would induce degradation of endogenous Stu2-AID, uncovering the phenotype of an ectopic mutant allele (*Miller et al., 2019*). We assessed in these strains the effects of the several Stu2 mutations on Stu2-kinetochore association as well as on cell viability. Kinetochore co-immunoprecipitations from *STU2-AID* cells harboring either hydrophobic-to-charged or alanine mutations of these residues, *stu2*$^{L869E,I873E,M876E}$ (*stu2*$^{EEE}$) or *stu2*$^{L869A,I873A,M876A}$ (*stu2*$^{AAA}$), showed loss of Stu2-kinetochore binding (*Figure 3B* and *Figure 3— figure supplement 1*). We also examined the localization of Stu2 in cells. In metaphase, Stu2 displays a characteristic bi-lobed distribution that largely overlaps with bioriented kinetochores clusters (*Aravamudhan et al., 2014*; *He et al., 2001*; *Humphrey et al., 2018*; *Kosco et al., 2001*). We examined *STU2-AID* cells arrested in metaphase (using a *CDC20-AID* allele) that carried either *STU2*$^{WT}$- or *stu2*$^{EEE}$-GFP fusion variants. Compared to *STU2*$^{WT}$, the *stu2*$^{EEE}$ mutant showed a dramatic loss in kinetochore proximal signal (*Figure 3C,D* and *Figure 3—figure supplement 2*), while displaying similar metaphase spindle length (measured by distance between Spc110-mCherry foci; *Figure 3E*). These same hydrophobic-to-charged or alanine mutations also resulted in a severe cell viability defect in the presence of auxin, exacerbated by addition of low concentrations of the microtubule destabilizing drug benomyl (*Figure 3F*). The individual *stu2* mutations also all had reduced cell viability (*Figure 3—figure supplements 3–5*). Thus, the hydrophobic residues in the CTS of Stu2, which contact Ndc80c in the crystal structure, are required for Stu2-kinetochore binding and cell viability in vivo.

The observed growth defect resulting from disruption of Stu2-kinetochore binding could, in principle, be due to loss of microtubule polymerase activity or due to association with other Stu2 CTS binding proteins (*Gunzelmann et al., 2018*; *Usui et al., 2003*; *Wolyniak et al., 2006*). To determine whether the viability defect of the most penetrant mutant, *stu2*$^{EEE}$, was due solely to loss of kinetochore binding or whether other Stu2 activities were involved, we tested rescue of the kinetochore binding of *stu2*$^{EEE}$ by rapamycin in cells expressing suitably engineered FRB and FKBP12 fusion proteins. We generated strains containing *STU2-AID* with *NUF2-FKBP12* at the endogenous *NUF2* locus and *STU2-FRB* alleles at an ectopic locus (*Figure 4—figure supplements 1–4*). We chose Nuf2 as the target for FKBP12 fusion because of the proximity of the C-termini of Nuf2 and Stu2 in the crystal structure. The cells also contained a mutant *TOR1* allele (*TOR1-1*) and *fpr1Δ*, to avoid inhibiting cell growth by the rapamycin treatment (*Haruki et al., 2008*). Cells expressing *stu2*$^{EEE}$-FRB were deficient for Stu2-kinetochore binding as expected. Expressing *NUF2-FKBP12* and adding rapamycin in culture rescued both the Stu2 binding defect (*Figure 3G*) and the cell viability defect of *stu2*$^{EEE}$ (*Figure 3H*). Because rapamycin addition rescued both defects, we infer that loss of kinetochore localization is the cause of viability loss. Thus, *stu2*$^{EEE}$ is a true separation of function mutation that allows us to assess the cellular consequences of disrupting Stu2-kinetochore association.

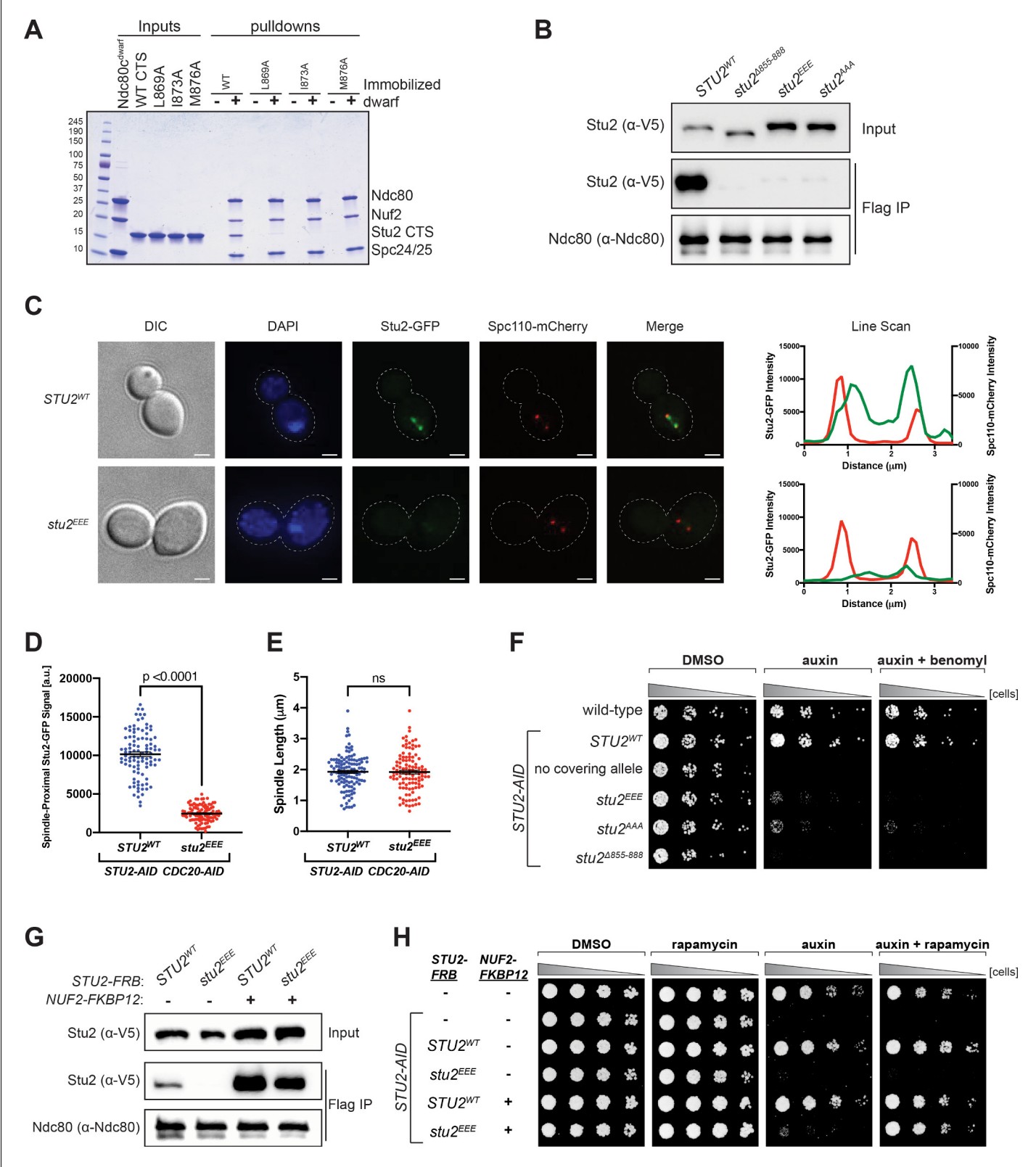

**Figure 3.** Effects of Stu2 mutations at the interface with Ndc80c on kinetochore association and cell viability and their rescue by re-tethering. (**A**) Effect of mutations at the binding interface. The Stu2 constructs contain the Stu2 coiled-coil domain, a glycine-serine linker, and either native or mutated C-terminal segment (CTS). Ndc80c^dwarf was immobilized on Co-NTA agarose, incubated with Stu2, washed, and eluted with 400 mM imidazole. (**B**) Exponentially growing *STU2-AID* cultures expressing an ectopic copy of *STU2* (*STU2^WT*, M622; *stu2^Δ855-888*, M653; *stu2^L869E,I873E,M876E* or *stu2^EEE*, M1444;

*Figure 3 continued on next page*

*Figure 3 continued*

stu2$^{L869A,I873A,M876A}$ or stu2$^{AAA}$, M1576) and also expressing from the genomic locus *DSN1-6His-3Flag* were treated with auxin 30 min prior to harvesting. Kinetochore particles were purified from lysates by anti-Flag immunoprecipitation (IP) and analyzed by immunoblotting. (C) Exponentially growing *STU2-AID CDC20-AID* cultures with an ectopically expressed *STU2-GFP* allele (*STU2$^{WT}$-GFP*, M1757; *stu2$^{EEE}$-GFP*, M1985) that also contained *SPC110-mCherry* (spindle pole) were treated with auxin for 2.5 hr to arrest cells in metaphase. Left: Representative micrographs for each are shown. White bars represent 2 µm. Right: Line scan plots through spindle axis (encompassing maximum Stu2-GFP signal) show Stu2-GFP and Spc110-mCherry intensity from example cells shown on left. (D) Spindle-Proximal Stu2-GFP signal from cells (n = 100) and line scan plots described in (C). Area under the curve for each line scan was measured using Fiji (p-value from a two-tailed unpaired t-test). (E) Spindle length (distance between Spc110-mCherry foci) was measured for cells (n = 100) described in (C) (p-value from a two-tailed unpaired t-test; n.s. = not significant). (F) Wild-type (M3), *STU2-AID* (no covering allele, M619), and *STU2-AID* cells expressing various *STU2-3V5* alleles from an ectopic locus (*STU2$^{WT}$*, M622; *stu2$^{EEE}$*, M1444; *stu2$^{AAA}$*, M1576; *stu2$^{\Delta855-888}$*, M653) were serially diluted (fivefold) and spotted on plates containing DMSO, 500 µM auxin, or 500 µM auxin + 5 µg/mL benomyl. (G) Exponentially growing *STU2-AID fpr1Δ TOR1-1* cultures expressing an ectopic copy of *STU2-FRB* with wild-type *NUF2* (*STU2$^{WT}$*, M1513; *stu2$^{EEE}$*, M1515) or with *NUF2-FKBP12* (*STU2$^{WT}$*, M1505; *stu2$^{EEE}$*, M1507) were treated with rapamycin and auxin 30 min prior to harvesting. Protein lysates prepared, subjected to α-Flag IP, and analyzed as in (B). (H) *fpr1Δ TOR1-1* cells (M1375), *STU2-AID fpr1Δ TOR1-1* cells (no covering allele, M1476), and *STU2-AID fpr1Δ TOR1-1* cells expressing *STU2-FRB* alleles at an ectopic locus with and without *NUF2-FKBP12* (M1513; M1515; M1505; M1507 – same genotypes as in D) were serially diluted (fivefold) and spotted on benomyl plates containing DMSO, 50 ng/mL rapamycin, 500 µM auxin, or 500 µM auxin + 50 ng/mL rapamycin.

The online version of this article includes the following figure supplement(s) for figure 3:

**Figure supplement 1.** Reproduction of western blot from *Figure 3B* also showing immunoblot signal for Stu2-3V5 with longer exposure included.
**Figure supplement 2.** Stu2-GFP expression levels examined by immunoblotting.
**Figure supplement 3.** Cell viability assays for single residue hydrophobic-to-charged mutations.
**Figure supplement 4.** Cell viability assays for single residue hydrophobic-to-alanine mutations.
**Figure supplement 5.** Heterozygous diploid dissection of different *stu2* mutant alleles.

## Proper chromosome biorientation and timely cell cycle progression require kinetochore-associated Stu2

The cellular phenotypes of the *stu2$^{EEE}$* mutant are consistent with a role for kinetochore-associated Stu2 in error correction. If so, cells defective in Stu2-kinetochore binding should also have defective chromosome biorientation. We assessed chromosome biorientation by using a methionine-repressible *CDC20* allele (*pMET-CDC20*) to arrest cells in metaphase that also carry a fluorescently marked centromere of chromosome III (*Straight et al., 1996*). Under these conditions, opposing spindle pulling forces cause bioriented sister chromatids to separate, appearing as two distinct GFP puncta (*Pearson et al., 2001*). The frequency of biorientation in *STU2$^{WT}$* cells was at the level usually found in this assay (44 ± 2%, *Figure 4A,B*), but it was significantly lower in *stu2$^{EEE}$* cells (16 ± 1%, p=0.0002). The presence of non-bioriented centromeres in the *stu2$^{EEE}$* mutant could also result from a lack of pulling forces on bioriented sister chromatids caused by reduced microtubule dynamics reported in loss of function *stu2* mutants (*Kosco et al., 2001*; *Pearson et al., 2003*). To examine this possibility, we monitored the axial position of the unseparated sister centromeres along the length of the metaphase spindle in *stu2$^{EEE}$* expressing cells. Bipolar attached centromeres are more likely to be near the spindle equator, whereas monopolar/syntelic centromeres will remain close to the spindle pole to which they are attached (*He et al., 2001*; *Huang and Huffaker, 2006*; *Skibbens et al., 1993*). We found that most non-bioriented sister centromeres were found proximal to one of the two spindle poles in both *STU2$^{WT}$* and *stu2$^{EEE}$* cells (*Figure 4—figure supplement 5*), suggesting that biorientation defects are caused by the persistence of monopolar/syntelic attachments and not by lack of microtubule pulling forces.

Cells defective in error correction depend on the spindle checkpoint to prevent chromosome missegregation and have a delayed metaphase-anaphase transition (*Biggins and Murray, 2001*; *Shonn et al., 2003*; *Stern and Murray, 2001*). We indeed found a significant synthetic growth defect of *stu2$^{EEE}$* with deletion of either of the spindle checkpoint components *MAD2* (*Figure 4—figure supplement 6*) or *MAD3* in the presence of low concentrations of benomyl (*Figure 4C*), likely due to severe aneuploidy in *stu2$^{EEE}$ mad2Δ* and *stu2$^{EEE}$ mad3Δ* cells. To examine the timing of cell cycle progression, we released cells with a functional spindle checkpoint from a G1 arrest and monitored nuclear divisions. Consistent with error correction defects, cells expressing *stu2$^{EEE}$-FRB* had greatly delayed anaphase onset, rescued by tethering Stu2$^{EEE}$-FRB to the kinetochore with *NUF2-FKBP12* and rapamycin (*Figure 4D* and *Figure 4—figure supplement 7*). Finally, we employed a

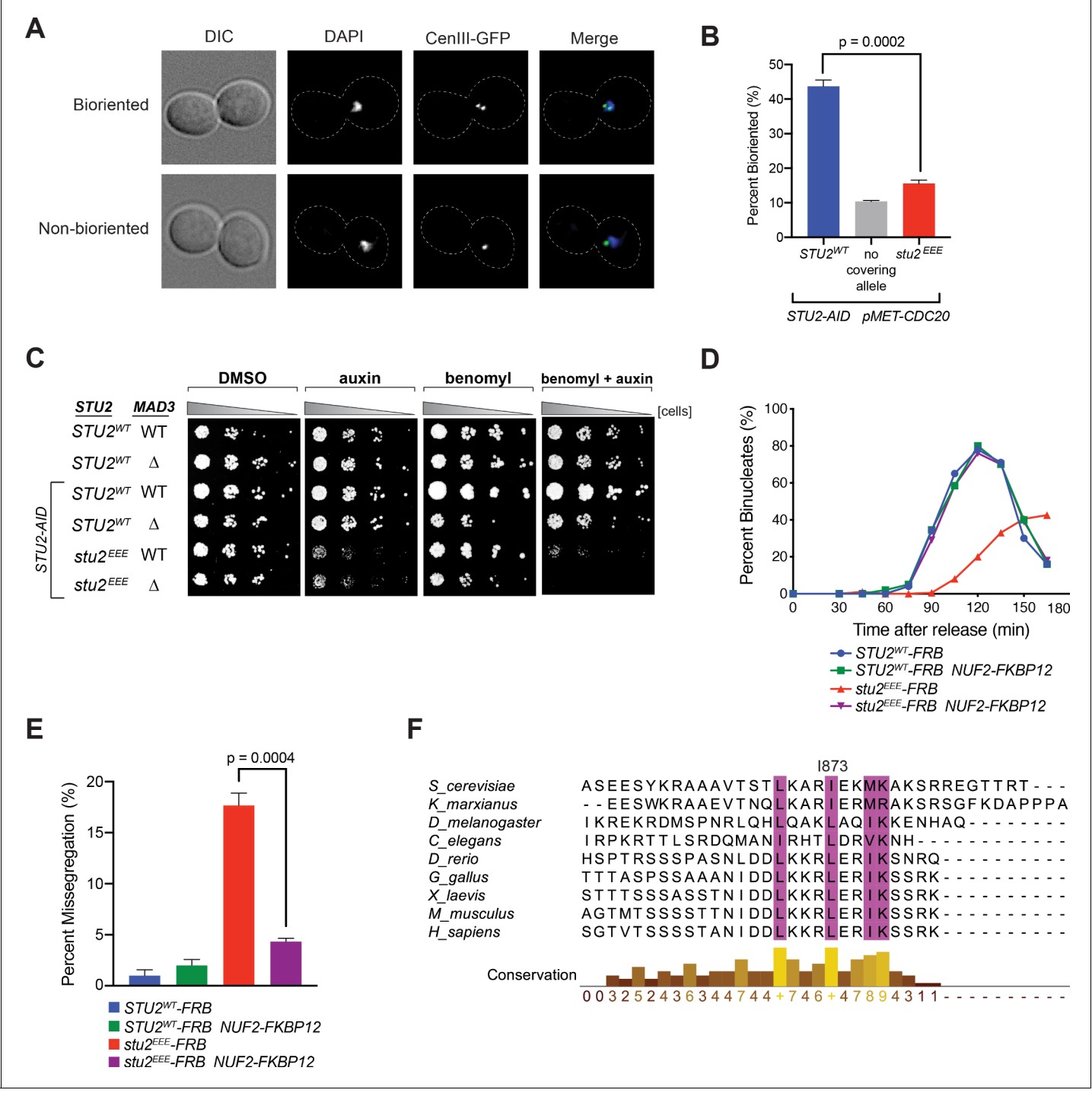

**Figure 4.** Cellular phenotypes of *stu2EEE* and conservation of key residues in multicellular eukaryotes. (**A**) Exponentially growing *STU2-AID pMET-CDC20* cultures with an ectopically expressed *STU2* allele (*STU2WT*, M1154; *stu2EEE*, M1610) or no ectopic allele (no covering allele, M1153) and also containing *CEN3* marked with GFP (*CEN3::lacO LacI-GFP*) were arrested in methionine + auxin containing media for 2.5 hr. Representative micrographs for bioriented and non-bioriented cells shown. (**B**) Percent bioriented cells were measured for cultures described in (**A**). Three replicates of n = 200 cells shown; p-value determined with unpaired t-test. (**C**) Wild-type cells (M3), cells with spindle checkpoint mutation (*mad3Δ*, M36), and *STU2-AID* cells expressing an ectopic copy of *STU2* without and with a spindle checkpoint mutation (*STU2WT*, M622; *STU2WT mad3Δ*, M1622; *stu2EEE*, M1444; *stu2EEE mad3Δ*, M1541) were serially diluted (fivefold) and spotted on plates containing DMSO, 500 µM auxin, 5 µg/mL benomyl, or 500 µM auxin + 5 µg/mL benomyl. (**D**) *STU2-AID* cells expressing *STU2-FRB* alleles at an ectopic locus with *NUF2* or *NUF2-FKBP12* (*STU2WT-FRB*, M1513; *stu2EEE-FRB*, M1515; *STU2WT-FRB NUF2-FKBP12*, M1505; *stu2EEE-FRB NUF2-FKBP12,* M1507) were released from a G1 arrest into auxin and rapamycin containing media. Cell cycle progression determined by the accumulation of binucleate cells. (**E**) *STU2-AID mad3Δ* cells expressing *STU2-FRB* alleles at an ectopic locus

*Figure 4 continued*

with *NUF2* or *NUF2-FKBP12* (*STU2^{WT}-FRB*, M2025; *stu2^{EEE}-FRB*, M2024; *STU2^{WT}-FRB NUF2-FKBP12*, M2027; *stu2^{EEE}-FRB NUF2-FKBP12*, M2026) that also contained a fluorescently labeled centromere of chromosome III were released from G1 arrest into auxin- and rapamycin-containing medium. Quantification of chromosome mis-segregation in anaphase (percent of binucleate cells with a fluorescently labeled chromosome III signal in only one of the two nuclei). Shown is an average of three biological replicates, n = 100 cells each. (F) Multiple sequence alignment of the Stu2 C-terminus and C-termini from Stu2 eukaryotic homologs. Histogram shows conservation score generated with Clustal Omega. Highly conserved residues, including hydrophobic amino acids important for Ndc80c binding, are boxed in purple.

The online version of this article includes the following figure supplement(s) for figure 4:

**Figure supplement 1.** Schematic illustrating re-tethering of Stu2^{EEE} to Ndc80c.
**Figure supplement 2.** Reproduction of western blot from *Figure 3D* also showing immunoblot signal for Dsn1-6His-3Flag.
**Figure supplement 3.** Characterization of *STU2-FRB* and *NUF2-FKBP12* fusion alleles.
**Figure supplement 4.** Tethering Stu2^{Δ855-888}-FRB to Ndc80c restores viability to the same degree as does tethering Stu2^{EEE}-FRB.
**Figure supplement 5.** Examining the axial position of unseparated sister centromeres along the length of the mitotic spindle.
**Figure supplement 6.** Synthetic growth defects of *stu2^{EEE}* with *mad2Δ* and *mad3Δ*.
**Figure supplement 7.** Replicate cell cycle delay experiment as described in *Figure 4D*.

similar tethering strategy to examine chromosome segregation fidelity. Cells carrying a fluorescently marked centromere of chromosome III and a deletion of *MAD3* (to ensure similar cell cycle progression) were released from a G1 arrest, and chromosome segregation was examined upon anaphase onset. Cells expressing *stu2^{EEE}-FRB* had marked errors in segregating chromosome III (18 ± 2%, *Figure 4E*), which were nearly always rescued by tethering Stu2^{EEE}-FRB to Nuf2-FKBP12 (4 ± 0.6%, *Figure 4E*). We conclude that correction of erroneous kinetochore-microtubule attachments in mitosis depends on kinetochore-bound Stu2. Moreover, rescue of the defect by ectopic tethering shows that CTS binding has a replaceable localization function and that the effector function for error correction lies elsewhere in the Stu2 molecule.

## Discussion

Previous studies have shown that Stu2, a microtubule plus-end polymerase, associates with kinetochores during mitosis in yeast cells (*He et al., 2001*; *Hsu and Toda, 2011*; *Kakui et al., 2013*; *Miller et al., 2016*; *Miller et al., 2019*; *Vasileva et al., 2017*) and contributes to the response to tension of kinetochores in vitro (*Miller et al., 2016*). The findings reported here indicate that kinetochore-bound Stu2 indeed contributes to error correction in response to the absence of tension in vivo. We determined the structure of a C-terminal peptide from Stu2 bound with a shortened construct of Ndc80c that preserves the junction between the two heterodimers it comprises (*Valverde et al., 2016*). We then used information from that structure to generate mutant yeast strains harboring *stu2* mutants defective in Ndc80c binding. Disrupting the Stu2:Ndc80c interaction led to diminished viability, aberrant chromosome biorientation, synthetic growth defects when combined with a spindle checkpoint mutant, and anaphase onset delays – all phenotypes consistent with defects in error correction. Chemically induced tethering of the mutant Stu2 to Ndc80c rescued all the defects we examined, including viability and cell cycle progression delays. Thus, the contributions of Stu2 to error correction in vivo depend on Ndc80c binding, not simply on its presence as a soluble factor or on its function as a microtubule polymerase. Our results do not rule out, however, that the required kinetochore localization of Stu2 couples its polymerase activity specifically with growth and shrinkage at the plus-ends of kinetochore microtubules during metaphase. Its polymerase activity might therefore also be essential. Stu2 association with Bik1 and Spc72 presumably couple this activity with other stages of spindle assembly or with other roles for microtubules in a yeast cell.

A published study of Stu2 dynamics at kinetochores using fluorescence recovery after photobleaching showed relatively high turnover, and its authors suggested that Stu2 might not be a stably associated kinetochore factor (*Aravamudhan et al., 2014*). The bi-lobed clusters of Stu2 in metaphase cells (*Goshima and Yanagida, 2000*; *He et al., 2001*; *Pearson et al., 2001*) probably contain both kinetochore-bound and purely microtubule tip-associated Stu2 pools, however, making definitive interpretation difficult. Our current data show that regardless of its exchange rate, association

of Stu2 with Ndc80c at the tetramer junction is necessary for its error correction function and that tethering Stu2 to the junction is sufficient to restore full activity of an association-defective mutant.

Conservation of the four-component organization of Ndc80c from yeast to mammals, despite no evidence for independent functions of either heterodimer, suggests a conserved role for the connection between them. The original plan when designing the Ndc80c$^{dwarf}$ construct was to provide a complex stiff enough to crystallize, while preserving the complete four-way junction. Several conserved features of the junction structure those crystals yielded, including exposed hydrophobic pockets and a skipped α-helical turn near the N-terminus of Spc24, indeed hinted that it might be a binding site for a conserved regulatory factor (*Valverde et al., 2016*). Stu2 is evidently at least one such factor. As now shown here, the Stu2 CTS adds a fifth helix to the four-helix bundle, without producing any conformational rearrangements, indicating that the role of the junction is simply to anchor Stu2 along the Ndc80c rod, rather than to receive an allosteric signal from it. That result does not rule out conformational switching by interactions between other parts of Stu2 thus anchored and other parts of the Ndc80c.

Bulky hydrophobic residues are present at key positions in the CTS of Stu2 orthologs from fungi to metazoans, and the entire CTS sequence has conserved features (*Figure 4F*). This conservation suggests that the orthologs also bind Ndc80c. Published work provides evidence for interaction of the *S. pombe* ortholog with the Ndc80 internal loop (*Hsu and Toda, 2011*; *Kakui et al., 2013*). The C-terminal position of the Ndc80c attachment segment of Stu2 and the length of the presumably flexible linker between it and the dimerizing coiled-coil would allow more distal parts of the protein to interact with almost any other site along the Ndc80c rod, including the Ndc80 internal loop and the globular head (*Miller et al., 2019*), and probably even allow the TOG domains to contact the curled plus-ends of an attached microtubule. These additional interactions are likely to be critical contributors to the mechanism by which Stu2 responds to tension or its absence and synergizes with the error-correcting detachment activity of Ipl1.

## Materials and methods

### Strain construction and microbial techniques

#### Yeast strains and plasmids

*Saccharomyces cerevisiae* strains used in this study, all derivatives of M3 (W303), are described in *Supplementary file 1*. Standard media and microbial techniques were used (*Sherman et al., 1974*). Yeast strains were constructed by standard genetic techniques. Construction of *pCUP1-GFP-LacI* and *ipl1-321* are described in *Biggins et al., 1999*, *CEN3::lacO:TRP1* is described in *Shonn et al., 2003*, *mad3Δ* in *Pinsky et al., 2006*, *DSN1-6His-3Flag* in *Akiyoshi et al., 2010*, *stu2-3V5-IAA7* in *Miller et al., 2016*, and *TOR1-1*, *fpr1Δ*, and *MPS1-FRB:KanMX* in *Aravamudhan et al., 2015*; *Haruki et al., 2008*. *STU2-FRB:HisMX* and *NUF2-FKBP12:HisMX* were constructed by PCR-based methods (*Longtine et al., 1998*). Strains containing the previously described *pMET-CDC20* allele were provided by Frank Uhlmann. *pGPD1-TIR1* integration plasmids (pM76 for integration at *HIS3* or pM78 for integration at *TRP1*) were provided by Leon Chan. Construction of a 3HA-IAA7 tagging plasmid (pM69) was described previously (*Miller et al., 2016*). Construction of a *LEU2* integrating plasmid containing wild-type *pSTU2-STU2-3V5* (pM225) and *pSTU2-stu2$^{Δ855–888}$-3V5* (pM267) are described in *Miller et al., 2016*; *Miller et al., 2019*. *STU2* variants were constructed by mutagenizing pM225 as described in *Liu and Naismith, 2008*; *Tseng et al., 2008*. Primers used in the construction of the above plasmids are listed in *Supplementary file 2*, and further details of plasmid construction including plasmid maps are available upon request.

### Auxin inducible degradation

The AID system was used essentially as described (*Nishimura et al., 2009*). Briefly, cells expressed C-terminal fusions of the protein of interest to an auxin responsive protein (IAA7) at the endogenous locus. Cells also expressed *TIR1*, which is required for auxin-induced degradation. 500 μM IAA (indole-3-acetic acid dissolved in DMSO; Sigma) was added to media to induce degradation of the AID-tagged protein. Auxin was added for 30 min prior to harvesting cells or as is indicated in figure legends.

## Spotting assay

For the spotting assay, the desired strains were grown for 2 days on plates containing yeast extract peptone plus 2% glucose (YPD) medium. Cells were then resuspended to OD600 ~1.0 from which a serial 1:5 dilution series was made and spotted on YPD+DMSO, YPD+500 µM IAA (indole-3-acetic acid dissolved in DMSO) or plates containing 3.5–5.0 µg/mL benomyl or 0.05 µg/mL rapamycin as indicated. Plates were incubated at 23°C for 2–3 days unless otherwise noted.

## Tetrad dissections

Diploid cells were sporulated, treated with 0.5 mg/mL zymolyase in 1 M sorbitol for 12 min, and tetrads were dissected on YPD plates using standard yeast microbial techniques. Plates were incubated at 23°C for 2 days.

## FRB/FKBP re-tethering

For re-tethering in culture, exponentially growing cultures were treated with 500 µM auxin and 0.05 µg/mL (55 nM rapamycin) 30 min prior to harvesting. For spotting assays on plates, 0.05 µg/mL rapamycin was used.

## Biorientation assay

In metaphase, sister kinetochores become bioriented and are transiently stretched apart by opposing microtubule pulling forces (*Goshima and Yanagida, 2000*; *He et al., 2001*; *Pearson et al., 2001*); the transient separation can be visualized by fluorescent marking of the centromere of a single chromosome (*Straight et al., 1996*). Biorientation was examined in metaphase arrested cells as judged by the fluorescently marked centromeres appearing as two distinct foci. Cells were grown in media lacking methionine (for *pMET-CDC20* containing strains). Exponentially growing cells were subsequently arrested in metaphase by the addition of 8 mM methionine each hour for 3 hr. An aliquot of cells was fixed with 3.7% formaldehyde in 100 mM phosphate buffer (pH 6.4) for 5 min. Cells were washed once with 100 mM phosphate (pH 6.4), resuspended in 100 mM phosphate, 1.2 M sorbitol buffer (pH 7.5) and permeabilized with 1% Triton X-100 stained with 1 µg/mL DAPI (4′, 6-diamidino-2-phenylindole; Molecular Probes).

## Fluorescence microscopy

Images were collected with a DeltaVison Elite wide-field microscope system (GE Healthcare) equipped with a scientific CMOS camera, using 60× objective (Olympus; NA = 1.42 PlanApoN) and immersion oil with a refractive index of n = 1.516. A Z-stack was acquired over a 2 µm width with 0.2 µm Z intervals. Images were deconvolved using the DeltaVision algorithm, maximally projected, and analyzed using the Fiji image processing package (ImageJ).

## Cell cycle progression and chromosome segregation assays

Cells were grown in YPD medium. Exponentially growing *MAT*a cells with or without a tandem array of lacO sequences integrated proximal to *CEN3* (*Shonn et al., 2003*) and a LacI-GFP fusion (*Biggins et al., 1999*; *Straight et al., 1996*) were arrested in G1 with 1 µg/mL α-factor. When arrest was complete, cells were released into medium lacking α-factor pheromone and containing 500 µM IAA and 0.05 µg/mL rapamycin at 23°C. For determining cell cycle progression, ~75 min after G1 release, 1 µg/mL α-factor was added to prevent a second cell division and samples were taken every 15 min after G1 release to determine cell cycle state (via nuclear morphology of DAPI stained nuclei). To assess chromosome segregation in cells containing GFP labeled *CEN3*, we sampled cells during anaphase and determined the percent of missegretated chromosomes.

## Protein biochemistry

### Purification of native kinetochore particles

Native kinetochore particles were purified from asynchronously growing *S. cerevisiae* cells as described below. Dsn1-6His-3Flag was immunoprecipitated with anti-Flag essentially as described in *Akiyoshi et al., 2010*. Cells were grown in yeast peptone dextrose (YPD) rich medium. For strains containing *STU2-AID*, cells were treated with 500 µM auxin 30 min prior to harvesting. Protein lysates were prepared by mechanically disrupted in the presence of lysis buffer using glass beads

and a beadbeater (Biospec Products). Lysed cells were resuspended in buffer H (BH) (25 mM HEPES pH 8.0), 2 mM MgCl2, 0.1 mM EDTA, 0.5 mM EGTA, 0.1% NP-40, 15% glycerol with 150 mM KCl containing protease inhibitors (at 20 µg/mL final concentration for each of leupeptin, pepstatin A, chymostatin, and 200 µM phenylmethylsulfonyl fluoride) and phosphatase inhibitors (0.1 mM Na-orthovanadate, 0.2 µM microcystin, 2 mM β-glycerophosphate, 1 mM Na pyrophosphate, 5 mM NaF) followed by centrifugation at 16,100 g for 30 min at 4°C to clarify lysate. Dynabeads conjugated with anti-Flag antibodies were incubated with extract for 3 hr with constant rotation, followed by three washes with BH containing protease inhibitors, phosphatase inhibitors, 2 mM dithiothreitol (DTT), and 150 mM KCl. Beads were further washed twice with BH containing 150 mM KCl and protease inhibitors. Associated proteins were eluted from the beads by boiling in 2× SDS sample buffer.

## Immunoblot analysis

For immunoblot analysis, cell lysates were prepared as described above or by pulverizing cells with glass beads in sodium dodecyl sulfate (SDS) buffer using a bead-beater (Biospec Products). Standard procedures for sodium dodecyl sulfate-polyacrylamide gel electrophoresis (SDS-PAGE) and immunoblotting were followed as described in *Burnette, 1981*; *Towbin et al., 1979*. A nitrocellulose membrane (Bio-Rad) was used to transfer proteins from polyacrylamide gels. Commercial antibodies used for immunoblotting were as follows: α-Flag, M2 (Sigma-Aldrich) 1:3000; α-V5 (Invitrogen) 1:5000. Antibodies to Ndc80 were a kind gift from Arshad Desai and used at anti-Ndc80 (OD4) 1:10,000. The secondary antibodies used were a sheep anti-mouse antibody conjugated to horseradish peroxidase (HRP) (GE Biosciences) at a 1:10,000 dilution or a donkey anti-rabbit antibody conjugated to HRP (GE Biosciences) at a 1:10,000 dilution. Antibodies were detected using the SuperSignal West Dura Chemiluminescent Substrate (Thermo Scientific).

## Recombinant protein expression and purification

The constructs used to express the Ndc80c$^{dwarf}$ are identical to those used previously (*Valverde et al., 2016*). Briefly, shortened versions of Ndc80/Nuf2 and Spc24/Spc25 were cloned into the orthogonal expression vectors pETduet1 and pRSFduet, respectively. Proteins were co-expressed in Rosetta 2(DE3) pLysS *E. coli* (Novagen). Cells were grown at 37°C in 2XYT media to an OD$_{600}$ of 0.8, induced with 200 µM IPTG, and incubated overnight with shaking at 225 RPM at 18°C. Cell pellets from 6 L of culture were resuspended in 150 mL of buffer containing 50 mM Tris pH 8.0, 250 mM NaCl, 10 mM imidazole pH 8.0, 5 mM β-mercaptoethanol, 1 mM PMSF, 1 µg/mL pepstatin, 1 µg/mL aprotinin, 1 µg/mL leupeptin, 83 µg/mL lysozyme, and 30 µg/mL DNase I. Cells were lysed by sonication, and the lysate was clarified by centrifugation and applied to Ni-NTA agarose equilibrated with buffer containing 20 mM Tris pH 8.0, 100 mM NaCl, 10 mM imidazole pH 8.0, and 5 mM β-mercaptoethanol (equilibration buffer). Bound material was washed with equilibration buffer containing 20 mM imidazole and 500 mM NaCl and then with equilibration buffer. Proteins were eluted with equilibration buffer containing 400 mM imidazole pH 8.0 and 50 mM NaCl. Eluates were treated overnight with TEV protease to remove the 6-His tag from the N-terminus of Ndc80 (except protein immobilized for pulldown experiments) and subjected to anion exchange chromatography using a Hi-trap Q HP (Cytiva), followed by size-exclusion chromatography using a Hi-load Superdex 200 16/60 column (Cytiva) equilibrated with 10 mM HEPES pH 7.0, 100 mM NaCl, and 2 mM tris(2-carboxyethyl)phosphine (TCEP).

Full length Stu2 was expressed in SF9 cells using the Bac-To-Bac system. Following infection, protein expression was allowed to proceed for 72 hr at 27°C. Cells were resuspended in 50 mM Tris pH 7.2, 500 mM NaCl, 10 mM imidazole pH 7.2, 5 mM β-mercaptoethanol, 0.1% TWEEN-20, 0.1% IPGAL, 1 mM PMSF, 1 µg/mL pepstatin, 1 µg/mL aprotinin, 1 µg/mL leupeptin, and 30 µg/mL DNase I. Cells were lysed using a Dounce homogenizer. The lysate was clarified by centrifugation and applied to Ni-NTA agarose equilibrated with 20 mM HEPES pH 7.2, 100 mM NaCl, 10 mM imidazole pH 7.2, and 5 mM β-mercaptoethanol. Resin was washed with buffer containing 500 mM NaCl and 20 mM imidazole, washed again with equilibration buffer and eluted with buffer containing 400 mM imidazole and 50 mM NaCl. Eluates were then subjected to cation exchange chromatography using a 5 mL Hi-Trap SP HP column (Cytiva). Fractions containing full-length Stu2 were applied to a Superose 6 10/300 column (Cytiva) equilibrated with 10 mM tris pH 7.5, 150 mM NaCl, and 2 mM TCEP.

The C-terminal fragments of Stu2 used in the pulldown experiments were cloned into a pLIC vectors with a TEV-cleavable N-terminal 6-His tag and transformed into Bl-21 AI cells (Invitrogen). Cells were grown to an $OD_{600}$ of 0.6 in TB media with 0.1% glucose and were induced by addition of IPTG and arabinose to 200 µM and 0.2%, respectively, incubated at 37°C for 4 hr to consume the glucose and then at 18° overnight. Cells were harvested by centrifugation, resuspended in 50 mM Tris pH 7.2, 250 mM NaCl, 10 mM imidazole pH 8.0, 5 mM β-mercaptoethanol, 1 mM PMSF, 1 µg/mL pepstatin, 1 µg/mL aprotinin, 1 µg/mL leupeptin, 83 µg/mL lysozyme, 30 µg/mL DNase, and lysed by sonication. The lysates were clarified by centrifugation and applied to Ni-NTA agarose equilibrated with 20 mM HEPES pH 7.2, 100 mM NaCl, 10 mM imidazole pH 7.2, and 5 mM β-mercaptoethanol. Eluates were treated with TEV protease overnight and then subjected to cation exchange chromatography on a 5 mL Hi-Trap SP HP column (Cytiva). Fractions containing Stu2 were then applied to a Superdex200 10/300 column (Cytiva) equilibrated with 10 mM HEPES pH 7.2, 150 mM NaCl, and 2 mM TCEP.

## Peptide

A 33-mer C-terminal Stu2 peptide, with a selenomethionine in place of the natural methionine at position 876 (EESYKRAAAVTSTLKARIEK(Mse)KAKSRREGTTRT), was synthesized at the Tufts University Core facility. The lyophilized peptide was resuspended to a final concentration of 10 mg/mL in a buffer containing 10 mM TRIS pH 7.0, 100 mM NaCl, and 2 mM TCEP.

## Crystallization, diffraction data collection, and structure determination

Ndc80c$^{dwarf}$ was concentrated to 15 mg/mL in an Amicon centrifugal concentrator in buffer containing 10 mM HEPES pH 7.0, 100 mM NaCl, and 2 mM TCEP. The Se-Met Stu2 peptide was diluted to 2 mg/mL in the same buffer. The protein was crystallized using hanging-drop vapor diffusion with a well solution containing 1.3 M ammonium sulfate and 0.1 mM HEPES pH 7.3. The hanging drops contained 1 µL each of the Ndc80c, peptide, and well solutions, resulting in an approximately threefold molar excess of peptide over Ndc80c$^{dwarf}$. Crystals were cryoprotected by a 3–5 min soak in well solution supplemented with 25% glucose and flash-frozen by plunging into liquid $N_2$. The complex crystallized in space group C222$_1$ (a = 190.89 Å, b = 183.3 Å, c = 124.32 Å). Data to a minimum Bragg spacing of 2.7 Å were recorded on the NE-CAT beamline 24-ID-C at the Advanced Photon Source, indexed and integrated using HKL2000, and scaled and merged using Scalepack as implemented in HKL2000 (*Otwinowski and Minor, 1997*; *Supplementary file 3*). The structure was determined by molecular replacement in Phenix (*Adams et al., 2010*), using the Ndc80c$^{dwarf}$ (PDB 5TCS) as a search model, yielding clear density for the Stu2 peptide. Model building was carried out in Coot (*Emsley et al., 2010*) and refinement, in Phenix (*Adams et al., 2010 Supplementary file 3*). We used the anomalous signal from a Se-methionine residue to calculate an anomalous difference map, confirming that the Se atom was at a position consistent with the chosen register. Coordinates and diffraction data have been deposited in the protein data bank, PDB ID: 7KDF.

## Pulldown experiments

For the pulldowns in *Figures 1D* and *3A*, Ndc80c$^{dwarf}$ with a 6-His affinity tag on the N-terminus of Ndc80 was immobilized to saturation on Co-NTA agarose by incubation with gentle agitation at 4°C. Beads were pelleted by centrifugation, washed three times in 20 mM HEPES pH 7.5, 200 mM NaCl, 2 mM imidazole pH 7.2, 2 mM β-mercaptoethanol, and 0.1% TWEEN-20 (wash buffer), and incubated with the relevant Stu2 construct for 30 min. Beads were again pelleted, washed three times with wash buffer, pelleted a final time, and the wash buffer aspirated from the tube. To elute bound proteins, 50 µL of wash buffer supplemented with 500 mM imidazole pH 7.5 was added to ~25 µL of beads. After an additional spin, the eluted proteins were visualized by SDS-PAGE. The pulldown in *Figure 1B* was carried out identically, with the exceptions that it used Ni-NTA agarose beads and that the wash buffer contained 150 mM instead of 200 mM NaCl.

## Affinity of Stu2 CTS and Ndc80c$^{dwarf}$

A Stu2 CTS peptide (855-888) with a C-terminal cysteine, synthesized by the Tufts University Core Facility, was labeled with Oregon Green maleimide (Thermo Fisher) according to the manufacturer's

instructions. The labeled peptide was separated from unreacted dye, first by cation exchange chromatography with Source 15S resin (Cytiva), followed by gel filtration chromatography on a Superdex 75 column (Cytiva). Fluorescence polarization was measured with a SpectraMax plate reader (Molecular Devices) at 25°C with excitation and emission wavelengths set at 484 and 525 nm, respectively. Binding of 50 nM Oregon Green CTS to 10 μM Ndc80c$^{dwarf}$ was carried out in buffer containing 10 mM HEPES pH 7.0 and 100 mM NaCl, adding increasing concentrations of unlabeled Stu2 CTS peptide. Measurements were carried out in duplicate. Curve fitting used a single site saturation binding model implemented in GraphPad Prism 9.

## Acknowledgements

We thank Arshad Desai for providing antibodies, Sue Biggins, Leon Chan, Ajit Joglekar, Frank Uhlmann, and Bri Stavaas for reagents, and to Sue Biggins for critical reading of the manuscript. The staff at the Northeastern Collaborative Access Team (NE-CAT) beamlines assisted with x-ray crystallographic data collection. NE-CAT is supported by NIH grant P30 GM124165, using resources of the Advanced Photon Source, operated by Argonne National Laboratory under Contract DE-AC02-06CH11357.

## Additional information

### Funding

| Funder | Grant reference number | Author |
|---|---|---|
| Damon Runyon Cancer Research Foundation | Dale F. Frey Award for Breakthrough Scientists.29-18 | Matthew P Miller |
| Howard Hughes Medical Institute | | Stephen C Harrison |

The funders had no role in study design, data collection and interpretation, or the decision to submit the work for publication.

### Author contributions

Jacob A Zahm, Michael G Stewart, Conceptualization, Formal analysis, Investigation, Methodology, Writing - original draft, Writing - review and editing; Joseph S Carrier, Conceptualization, Investigation, Writing - review and editing; Stephen C Harrison, Matthew P Miller, Conceptualization, Formal analysis, Supervision, Funding acquisition, Writing - original draft, Writing - review and editing

### Author ORCIDs

Jacob A Zahm https://orcid.org/0000-0002-9600-5433
Michael G Stewart https://orcid.org/0000-0001-6581-6332
Stephen C Harrison https://orcid.org/0000-0001-7215-9393
Matthew P Miller https://orcid.org/0000-0003-2012-7546

### Decision letter and Author response

Decision letter https://doi.org/10.7554/eLife.65389.sa1
Author response https://doi.org/10.7554/eLife.65389.sa2

## Additional files

### Supplementary files

• Supplementary file 1. Strains used in this study. All strains are derivatives of M3 (W303).

• Supplementary file 2. Plasmids and primers used in this study. All pM plasmids (except pM646) are derivatives of pM225.

• Supplementary file 3. Data collection and refinement statistics.

- Transparent reporting form

## Data availability

Diffraction data have been deposited in PDB under the accession code 7KDF.

The following dataset was generated:

| Author(s) | Year | Dataset title | Dataset URL | Database and Identifier |
|---|---|---|---|---|
| Zahm JA, Stewart MG, Miller MP, Harrison SC | 2020 | Structure of Stu2 Bound to dwarf Ndc80c | https://www.rcsb.org/structure/7KDF | RCSB Protein Data Bank, 7KDF |

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
