## [Decision Letter]

**Acceptance summary:**

In this manuscript, Zahm and co-workers shed light on a binding mechanism of great relevance for chromosome bi-orientation and genomic stability. Using X-ray crystallography, the authors describe the structure of a binding interface between a conserved microtubule-binding protein, Stu2/chTOG/xMAP215, and the NDC80 complex, the primary kinetochore-microtubule attachment factor in eukaryotic cells. The authors unveil how Stu2 is recruited to kinetochores via the NDC80 complex, and they harness this structure to generate a separation of function mutant that demonstrates the absolute relevance of kinetochore-associated Stu2 for chromosome bi-orientation. The study provides a solid basis for further analyses of how tension sensing at the kinetochore promotes bi-orientation.

**Decision letter after peer review:**

Thank you for submitting your article "Structural Basis of Stu2 Recruitment to Yeast Kinetochores" for consideration by *eLife*. Your article has been reviewed by three peer reviewers, and the evaluation has been overseen by a Reviewing Editor and Kevin Struhl as the Senior Editor. The following individuals involved in review of your submission have agreed to reveal their identity: Andrea Musacchio (Reviewer #1); Stefan Westermann (Reviewer #2).

Essential revisions

The reviewers and editors discussed the following experimental revision that the authors should address before publication.

For the experiment in Figure 4A, a spindle pole marker should be included. The lower percentage of split signals could either be generated by monopolar/syntelic attachments or the lack of pulling forces on bioriented chromosomes. A spindle pole marker will be useful to determine if the signal lies between poles or at one of the two poles to distinguish between these models, and also to ensure that metaphase spindle length is not affected by the Stu2 mutant.

The two additional essential revisions listed below should be addressable with added discussion in the text:

1) The authors should discuss the ramifications of addition of the peptide into the cryo-protectant solution. Reduction of peptide binding after the addition of the cryoprotectant could also explain "the weak peptide features on the map" if the peptide diffused away upon bringing the crystal into cryo-protectant solution.

2) The authors should add some discussion regarding the finding that the C-terminal extension of the CTS domain has density in the structure even though it does not have secondary structure. Does this region contribute to a crystal contact, and if so, are the authors confident of this structure in the protein?

---

## [Author Response]

Essential revisionsThe reviewers and editors discussed the following experimental revision that the authors should address before publication.For the experiment in Figure 4A, a spindle pole marker should be included. The lower percentage of split signals could either be generated by monopolar/syntelic attachments or the lack of pulling forces on bioriented chromosomes. A spindle pole marker will be useful to determine if the signal lies between poles or at one of the two poles to distinguish between these models, and also to ensure that metaphase spindle length is not affected by the Stu2 mutant.

Thank you to the reviewers for suggesting this experiment. We completely agree that distinguishing monopolar/syntelic attachments versus a lack of pulling forces on bioriented chromosomes is an important point, as well as ensuring that the *stu2* mutant does not affect mitotic spindle length. To address these points, we conducted two further experiments:

1) Figure 3E. We examined spindle length in metaphase arrested cells for wild-type and *stu2^EEE^* mutant cells and found them to be indistinguishable. In this experiment, we also examined the localization of GFP-fusion wild-type and mutant variants (Figure 3C-D). Consistent with the biochemical experiments in our original manuscript, we observed defects in kinetochore-proximal GFP signal in the mutant cells.

2) Figure 4—figure supplement 5. We monitored the axial position of the unseparated sister centromeres along the length of the metaphase spindle in *stu2^EEE^* expressing cells as suggested by the reviewers. We found that non-bioriented sister centromeres were predominantly found proximal to one of the two spindle poles in both *STU2^WT^* and *stu2^EEE^* cells, suggesting that biorientation defects are caused by the persistence of monopolar/syntelic attachments and not a lack of pulling forces. (see Figure 4—figure supplement 5 and accompanying in text)

The two additional essential revisions listed below should be addressable with added discussion in the text:1) The authors should discuss the ramifications of addition of the peptide into the cryo-protectant solution. Reduction of peptide binding after the addition of the cryoprotectant could also explain "the weak peptide features on the map" if the peptide diffused away upon bringing the crystal into cryo-protectant solution.

This is a very good point, please see discussion below.

2) The authors should add some discussion regarding the finding that the C-terminal extension of the CTS domain has density in the structure even though it does not have secondary structure. Does this region contribute to a crystal contact, and if so, are the authors confident of this structure in the protein?

We thank reviewer #3 for noticing an inconsistency, due in part to a misunderstanding on our part about occupancy when drafting the manuscript initially. Our previous statement of lower occupancy was incorrect, as we realized when examining carefully the refinement results, but we failed to eliminate it in the submitted version. We have measured the Kd of the CTS peptide for Ndc80c^dwarf^ by competition fluorescence polarization as 48 μM, consistent with the requirement for a dimer interaction in vitro and with the likely requirement for additional, cooperative interactions in vivo. The adventitious interaction of the C-terminal segment with a neighboring molecule in the crystal lattice probably accounts not only for retention of the peptide curing the ca. 30 minute cryo-soak, but also for the non-isomorphism (but same space group) of the present crystals with those of the unliganded Ndc80c^dwarf^ and for the somewhat different crystallization conditions. We have added a panel to Figure 1 (Figure 1E) showing the Kd measurement and a section to Materials and methods describing it. We have also added the following two sentences to the text when describing the structure:

“This lattice interaction may account for the somewhat different unit-cell dimensions in our crystals from those of the unliganded Ndc80c^dwarf^ in the same space group (Valverde et al., 2016), as well as for the nearly full occupancy but slightly higher local thermal parameters, as determined by refinement, for the low affinity peptide, which did not diffuse out during the short soak in cryopreservative (see Materials and methods).”